# Plant MIR167e-5p Inhibits Enterocyte Proliferation by Targeting β-Catenin

**DOI:** 10.3390/cells8111385

**Published:** 2019-11-04

**Authors:** Meng Li, Ting Chen, Jia-Jian He, Jia-Han Wu, Jun-Yi Luo, Rui-Song Ye, Mei-Ying Xie, Hao-Jie Zhang, Bin Zeng, Jie Liu, Qian-Yun Xi, Qing-Yan Jiang, Jia-Jie Sun, Yong-Liang Zhang

**Affiliations:** Guangdong Provincial Key Laboratory of Animal Nutritional Control, National Engineering Research Center for Breeding Swine Industry, College of Animal Science, South China Agricultural University, Guangzhou 510642, China; limengscau@163.com (M.L.); allinchen@scau.edu.cn (T.C.); scauhejiajian@163.com (J.-J.H.); jiahan94@foxmail.com (J.-H.W.); luojunyi@scau.edu.cn (J.-Y.L.); yiguangnian2004@163.com (R.-S.Y.); xxiemy@163.com (M.-Y.X.); zhanghj089@126.com (H.-J.Z.); zhanghj089@126.com (B.Z.); 15800202206@163.com (J.L.); xqy0228@163.com (Q.-Y.X.); qyjiang@scau.edu.cn (Q.-Y.J.)

**Keywords:** MIR167e-5p, β-catenin, enterocyte, proliferation

## Abstract

MicroRNAs (miRNAs) are important negative regulators of genes involved in physiological and pathological processes in plants and animals. It is worth exploring whether plant miRNAs play a cross-kingdom regulatory role in animals. Herein, we found that plant MIR167e-5p regulates the proliferation of enterocytes in vitro. A porcine jejunum epithelial cell line (IPEC-J2) and a human colon carcinoma cell line (Caco-2) were treated with 0, 10, 20, and 40 pmol of synthetic 2′-*O*-methylated plant MIR167e-5p, followed by a treatment with 20 pmol of MIR167e-5p for 0, 24, 48, and 72 h. The cells were counted, and IPEC-J2 cell viability was determined by the MTT and EdU assays at different time points. The results showed that MIR167e-5p significantly inhibited the proliferation of enterocytes in a dose- and time-dependent manner. Bioinformatics prediction and a luciferase reporter assay indicated that MIR167e-5p targets β-catenin. In IPEC-J2 and Caco-2 cells, MIR167e-5p suppressed proliferation by downregulating β-catenin mRNA and protein levels. MIR167e-5p relieved this inhibition. Similar results were achieved for the β-catenin downstream target gene c-Myc and the proliferation-associated gene PCNA. This research demonstrates that plant MIR167e-5p can inhibit enterocyte proliferation by targeting the β-catenin pathway. More importantly, plant miRNAs may be a new class of bioactive molecules for epigenetic regulation in humans and animals.

## 1. Introduction

MicroRNAs (miRNAs) are critical biological molecules and have attracted much attention for more than a decade. The functions of miRNA are well established in plants and animals. These single-stranded molecules impact on fundamental biological processes, including proliferation, differentiation, immune responses, and metabolism in different species. The recent detection of plant miRNAs in body fluids, including serum [1], urine [2], saliva [3], and milk [4], suggests that these endogenous circulating miRNAs may be broadly implicated in miRNA-mediated control of gene expression. Subsequent studies have suggested that plant miRNAs might regulate gene expression in mammalian cells. In mouse liver, plant MIR168a targets low-density lipoprotein receptor adapter protein 1 (LDLRAP1), which results in the elevation of LDL-cholesterol level in the plasma [5]. Plant MIR2911 has been reported to target the *PB2* and *NS1* genes encoded by influenza A viruses (IAVs) H1N1, inhibiting H1N1 viral replication [6]. MIR159 is capable of inhibiting cancer proliferation by targeting T cell factor (TCF) 7 [7]. MIR156a can directly target junction adhesion molecule-A (JAM-A), reducing inflammatory cytokine-induced monocytes adhesion [8]. Plant MIR162a in beebread can even regulate honeybee caste development [9]. Surprisingly, a recent study shows that *Cuscuta campestris* miRNAs were induced through the haustorium to target and reduce the accumulation of mRNAs, such as those for F-box/RNI-like superfamily protein (TIR1) and sieve element occlusion amino-terminus protein 1 (SEOR1), in host plants [10]. These findings provide new evidence that food-derived miRNAs may elicit the regulation of gene expression and be used in nutritional therapies for animal and human beings.

Some researchers have identified all intestinal miRNAs and shown that miRNAs play a vital role in the differentiation and function of the intestine by using gene ablation of Dicer1 [11]. A study demonstrated that grape exosome-like nanoparticles (GELNs) could penetrate the intestinal mucus barrier and were taken up by mouse intestinal stem cells, causing a significant induction of Lgr5 intestinal stem cells through the Wnt/β-catenin pathway and protecting the mice from dextran sulphate sodium (DSS)-induced colitis [12]. Subsequently, it was found that edible plant-derived exosome-like nanoparticles (EPDELNs) could instigate EPDELN-mediated interspecies communication by inducing the of genes for anti-inflammatory cytokines, antioxidation, and activation of Wnt signaling, which are crucial for maintaining intestinal homeostasis [13]. Many EPDELNs have anti-inflammatory properties. MiRNAs have been detected in 11 EPDELNs and found to have the potential to regulate human mRNA [14]. Exosome-like nanoparticles (ELNs) mdo-miR7267-3p-mediated targeting of *Lactobacillus rhamnosus* (LGG) monooxygenase ycnE yielded increased indole-3-carboxaldehyde, inducing the production of interleukin (IL)-22 and ameliorating mouse colitis via IL-22-dependent mechanisms [15]. MiRNAs can be selectively encapsulated in extracellular vesicles and perform anti-inflammatory functions. These findings suggest that plant miRNAs have important effects on the regulation of intestinal functions.

The intestinal tract not only is the first line of defense against harmful pathogens, but also is the main place for the digestion and absorption of nutrients. Ontogenetic development of the mammalian intestine is a topologically and temporally highly organized process. Intestinal stem cells (ISCs) constantly produce enterocytes and endocrine cells to maintain homeostasis in the intestinal epithelium [16]. One of the crucial results of this process is the formation of a specialized intestinal epithelium that exerts digestive and absorptive functions, as well as certain endocrine and immunological roles [16]. The survival of newborns and the maintenance of homeostasis require the regulation of the intestinal epithelium operating as a crucial unit [17]. Therefore, it is particularly important to maintain normal intestinal morphology and structure. The morphogenetic process and the acquisition of particular cell fates are coordinated by a relatively small number of highly evolutionarily conserved signaling pathways, including the canonical Wnt signaling pathway [18]. The standard model of Wnt/β-catenin signaling states that, upon stimulation by the Wnt ligand, β-catenin accumulates and subsequently translocates to the nucleus to activate the TCF-dependent transcription of a variety of target genes, including cellular myelocytomatosis oncogene (c-Myc) [19]. This pathway has been shown to play a central role in cell proliferation, differentiation, and stem cell maintenance [18]. Sun et al. found that miR-320a can target the β-catenin gene and inhibit the proliferation of colorectal cancer cells [20]. While plant miRNAs have been involved in canonical Wnt activity regulatory processes, thier effects on the proliferation of intestinal cells remain to be explored.

We hypothesized that plant miRNA might be an exogenous regulator influencing animal enterocytes’ functions. We found that plant MIR167e-5p could inhibit enterocyte proliferation and further identified β-catenin as a target of MIR167e-5p. This study demonstrates that plant miRNAs can influence the growth of enterocytes and thus serve as important bioactive molecules.

## 2. Materials and Methods

### 2.1. Cell Culture and Transfection of Cells with MIR167e-5p

IPEC-J2, STC-1, and Caco-2 cells were seeded at a density of 1 × 10^5^, 4 × 10^5^, and 2 × 10^5^ cells, respectively, in 12-well plates overnight (70–80% confluency) and transfected for 6 hours (h) the following day using Lipofectamine 2000 (Invitrogen, Carlsbad, CA, USA), according to the manufacturer’s instructions. The treatment groups were as follow: MIR167e-5p group (MIR167e-5p mimics, 20 pmol), NC group (normal control, 20 pmol), MIR167e-5p inhibitor group (MIR167e-5p inhibitor, 20 pmol), i-NC group (inhibitor normal control, 20 pmol,), MIR167e-5p+inhibitor group (20 pmol MIR167e-5p mimics +40 pmol MIR167e-5p inhibitor). The cells were harvested at 0, 24, 48, or 72h after transfection for cell counting, real-time quantitative PCR (qPCR), and western blot analysis.

### 2.2. RNA Isolation

Total RNA was extracted from the cells using TRIzol Reagent (Invitrogen) according to the manufactures’ instruction. RNA concentration was determined by an ND-2000 Nanodrop spectrophotometer (NanoDrop Technology, Wilmington, DE, USA). RNA integrity was examined by gel electrophoresis using a 0.8% agarose gel. The RNA was used for further PCR and qPCR analyses.

### 2.3. PCR and qPCR Analyses

Total RNA (2 μg) was reverse-transcribed to cDNA using M-MLV-R transcriptase (Promega, Madison, WI, USA) with OligodT18 (for mRNA) or a specific stem-loop RT primer (for miRNA). The samples were amplified by PCR using complementary primers. To quantify miRNAs and mRNAs, cDNA was diluted 5-fold with ddH_2_O, and qPCR (20 μL) was performed on a Bio-Rad CFX Manager 3.1 instrument (Applied Biosystems, Waltham, MA, USA) with U6 (miRNA control) or β-actin (mRNA control) as internal controls. The reactions contained 2 μL cDNA, 10 μL 2× GoTaq Qpcr Master Mix (Promega), and 0.4 μL of each primer (10 μM). The thermal profile of real-time PCR involved an initial denaturation step at 95 °C for 5 min (min), followed by 40 cycles at 94 °C for 15 s, 15 s at the corresponding annealing temperature (Tm), extension at 72 °C for 30 s, followed by a quick denaturation at 95 °C for 10 s, plus a slow ramp from Tm to 95 °C to generate a melting curve to confirm the specificity of the amplified product. A no template (NT) negative control was included for each miRNA and mRNA, and all reactions were performed in triplicate. The 2^−ΔCt^ method was employed to quantify and normalize the expression data. The miRNAs, mRNAs primers, and specific stem-loop RT primers were designed with Premier Primer 5.0 (Premier Software, San Jose, CA, USA) (Table 1).

### 2.4. Western Blot Analysis

Total proteins were isolated from the cells using RIPA buffer with 1 nM phenylmethylsulphonyl fluoride (PMSF) and quantified with a BCA Total Protein Assay Kit (Thermo Fisher, Waltham, MA, USA). The protein suspension (30 μg) was supplemented with β-mercaptoethanol, boiled for 10 min, resolved by 5% and 10% sodium dodecyl sulphate-polyacrylamide gel electrophoresis (SDS-PAGE), and transferred to a polyvinylidene difluoride membrane. β-catenin and c-Myc were detected by incubation with primary anti-human β-catenin (1:500, Sangon Biotech, Shanghai, China) and anti-human c-Myc (1:1000, Sangon Biotech) antibodies, followed by incubation with rabbit HRP-linked IgG1 secondary antibodies (1:50,000, Bioworld Technology, St Louis Park, MN, USA). β-actin (1:5000, Bioworld Technology) served as a control. Band intensities were measured using Image J and were normalized against β-actin band intensity.

### 2.5. Dual-Luciferase Reporter Assay

On the basis of the predicted miRNA–mRNAs binding sequences, normal sequences bearing an MIR167e-5p target site or with MIR167e-5p target site-deleted β-catenin 3′-UTR were generated by two complementary chemically synthesized primers (Sangon Biotech) as follows: wt-β-catenin-3′UTR-sense (tcgagttgtatctaaagtccggtgttgccagcttcagttggttcctgt), wt-β-catenin-3′UTR-antisense (ctagaggaaccaactgaagctggcaacaccggactttagatacaac), del-β-catenin-3′UTR-sense (tcgagttgtatctaaagtccggtgttgcagttggttcctgt), and del-β-catenin-3′UTR-antisense (ctagacaggaaccaactgcaacaccggactttagatacaac). The complementary oligonucleotides were re-suspended at a 1:1 ratio (1 μg/μL each) in annealing buffer (10 mM TRIS, pH 7.5–8.0, 50 mM NaCl, 1 mM EDTA) and heated at 95 °C for 10 min to denature the secondary structures. The temperature was then gradually lowered to room temperature. Annealed products were cloned into the pmir-GLO vector (Promega) downstream from the firefly luciferase coding region (between *XhoI* and *Xba*I sites). HeLa cells were seeded in 96-well cell culture plates (3.5 × 10^4^ cells per well) and cultured in RPMI 1640 (Life Technologies, Grand Island, NY, USA) with 10% fetal bovine serum (FBS). The next day, the cells were transfected with recombinant pmirGLO-3′UTR vector (100 ng/well) mixed with their corresponding MIR167e-5p mimics or NC (3 pmol/well, RiboBio, Guangzhou, Guangdong, China) for 6 h using Lipofectamine 2000 (Invitrogen). The cells were harvested 24 h after transfection, and luciferase activity was detected by a dual luciferase reporter assay system (Promega) according to the manufacturer′s recommendations. The normalized firefly luciferase activity (firefly luciferase activity/Renilla luciferase activity) of each construct was compared with that of the pmirGLO vector.

### 2.6. 3-(4,5-Dimethyl-2-thiazolyl)-2,5-diphenyl-2H-tetrazolium bromide (MTT) Assay

The MTT kit was purchased from Beyotime Biotechnology (Shanghai, China) and used according to the manufacturer’s protocol. Briefly, IPEC-J2 cells were seeded in 96-well plates at a density of 4000 cells per well with 200 μL of complete culture medium. After being allowed to adhere and spread for 12 h, the cells were treated with 3 pmol of MIR167e-5p, NC, inhibitor, and i-NC for 0, 24, 48, and 72 h. MTT assays were performed by incubating the treated IPEC-J2 cells with 20 μL (5 mg/mL) of MTT labelling solution. After 4 h of incubation, IPEC-J2 were lysed with 150 μL of DMSO, and the purple formazan crystals were solubilized for detection at 570 nm.

### 2.7. 5-Ethynyl-2′-deoxyuridine (EdU) Incorporation Assay

Cell proliferation was assessed by a Cell-Light EdU Apollo 488 In Vitro Imaging Kit (Ribobio), according to the manufacturer’s instructions. IPEC-J2 cells were incubated with 1:1000 EdU diluted in culture solution for 2 h at 37 °C before harvesting. After several washes, the cells were fixed with 4% paraformaldehyde (pH 7.4) for 30 min and incubated with glycine (2 mg/mL) for 5 min. The fixed cells were washed with 0.5% Triton-100 (diluted in phosphate-buffered saline (PBS)) for 10 min, followed by staining with Apollo buffer (RiboBio) for 30 min at room temperature in the dark. The cells were incubated with Hoechst 33342 dye at room temperature for 30 min in the dark. Finally, images were captured using fluorescent microscopy and IR–DIC imaging on an Eclipse FN-1 upright microscope (Nikon, Tokyo, Japan) at a magnification of 100×.

### 2.8. Statistical Analyses

Least significant difference (LSD) tests were used for multiple comparisons, and *t*-tests were employed for pairwise comparisons (*p* < 0.05 was considered statistically significant). Data are presented as the mean ± standard error (S.E.), and all analyses were performed using GraphPad Prism software 6.0 (GraphPad Software, San Diego, Canada).

## 3. Results

### 3.1. Synthetic MIR167e-5p Suppresses the Proliferation of Enterocytes

Recent studies indicated that exosomes from plants can protect mice from colitis [12], and miRNAs can be packaged into ELNs to regulate the intestinal microbiota [13,15]. Therefore, we hypothesized that plant miRNAs might also regulate mammalian enterocytes. To explore this idea, three species of intestinal cells were selected for experiments. Porcine jejunum epithelial cells IPEC-J2, mouse endocrine cells STC-1, and human colon carcinoma cells Caco-2 were transfected with either synthetic 2′-*O*-methylated plant MIR167e-5p or 2′-*O*-methylated control nucleic acids (NC). The cell number in each sample was calculated 24 h after treatment with 0, 10, 20, and 40 pmol MIR167e-5p. We observed that 10 pmol MIR167e-5p determined a remarkable decrease of IPEC-J2 (*p <* 0.01, Figure 1A), STC-1 (*p <* 0.01, Figure 1B), and Caco-2 (*p <* 0.01, Figure 1C) cells. Furthermore, to test the time-dependence of MIR167e-5p effect, the cells were treated with 20 pmol MIR167e-5p, and the cell number was determined after 0, 24, 48, and 72 h of treatment. The results showed an obvious decrease of IPEC-J2 cell number after 24 h (8.1%), 48 h (24.8%), and 72 h (24.6%) of treatment, compared to NC cells (Figure 1D). STC-1 cell number also decreased after 24 h (8.2%) and 48 h (10.9%) of treatment, but no significant difference at 72 h was found, compared to NC cells (Figure 1E). Similarly, Caco-2 cell number also decreased after 24 h (18.7%), 48 h (11.2%), and 72 h (17.0%) of treatment, compared to NC cells (Figure 1F). These results indicate that MIR167e-5p significantly inhibited the proliferation of intestinal cells. The MTT assay and the EdU incorporation assay were performed to further confirm the inhibition of MIR167e-5p. The MTT assay showed that the proliferation of IPEC-J2 cells was significantly suppressed after 72 h by MIR167e-5p, while MIR167e-5p inhibitor did not show any effect (Figure 1G). The EdU incorporation assay showed that the fluorescence intensity of IPEC-J2 cells was significantly reduced after both 24 and 48 h of MIR167e-5p treatment but not after treatment with MIR167e-5p inhibitor (Figure 1H,I), indicating significantly reduced DNA replication activity. Taken together, these results indicate that MIR167e-5p is capable of inhibiting enterocyte proliferation.

### 3.2. MIR167e-5p Targets the Transcript of β-Catenin

MIR167e-5p regulates various aspects of development in plants by targeting the auxin response factors ARF6 and ARF8 [21]. To identify targets of MIR167e-5p in mammals, miRanda was employed, and the results showed that MIR167e-5p might target β-catenin, a key molecule in the Wnt/β-catenin pathway. A putative binding site conserved among various species was located in β-catenin (Figure 2A). To confirm this relationship, a β-catenin partially normal 3′-UTR sequence and a sequence in which the MIR167e-5p binding site was deleted were cloned into a luciferase reporter plasmid (Figure 2A). HeLa cells were transfected with the reporter plasmids along with synthetic MIR167e-5p mimics or NC. The results showed that MIR167e-5p significantly reduced luciferase activity, and deletion of the MIR167e-5p binding site diminished this reduction (Figure 2B), which was preliminary evidence of their binding.

### 3.3. MIR167e-5p Inhibits the β-Catenin Pathway

To determine whether MIR167e-5p inhibits the β-catenin pathway, two enterocyte cell lines, IPEC-J2 and Caco-2 cells, were transfected with synthetic MIR167e-5p. Interestingly, the transfection led to a remarkable increase of MIR167e-5p in the cells, and MIR167e-5p inhibitor restored MIR167e-5p levels to the levels in NC cells (Figure 3A,E). More importantly, as the potential target of MIR167e-5p, β-catenin mRNA level was decreased after MIR167e-5p transfection compared to NC transfection (Figure 3B) in IPEC-J2 cells, indicating that MIR167e-5p probably enhanced the degradation of β-catenin mRNA. As a proliferative marker gene for intestine cell, PCNA mRNA examined by qPCR was significantly reduced by MIR167e-5p compared to NC and restored to a normal level by MIR167e-5p inhibitor (Figure 3C). We also examined c-Myc, a downstream molecule of β-catenin. Analysis by qPCR revealed that MIR167e-5p significantly reduced c-Myc mRNA level compared to NC, and this reduction was prevented by MIR167e-5p inhibitor (Figure 3D). Similarly, these three mRNAs were significantly reduced by MIR167e-5p in Caco-2 cells (Figure 3E–H), and β-catenin mRNA level was recovered to that in NC cells by MIR167e-5p inhibitor. These results indicate that MIR167e-5p may regulate the β-catenin pathway.

To further confirm the inhibition of MIR167e-5p, β-catenin and c-Myc proteins were examined by western blot. As shown in the results, MIR167e-5p seemed to downregulate β-catenin. β-catenin protein levels in IPEC-J2 cells were reduced, and this reduction could be restored by the inhibitor (Figure 4A,B). C-Myc was downregulated too, as shown in Figure 4A,C and recovered in the presence of the inhibitor. A similar inhibitory effect of MIR167e-5p was observed in Caco-2 cells (Figure 4D–F). These results provided further evidence that MIR167e-5p may inhibit the β-catenin pathway.

## 4. Discussion

MiRNAs are important negative regulators of genes involved in physiological and pathological processes in plants and animals. Whether plant miRNAs play a cross-kingdom regulatory role in animals is still disputed. Animal and plant miRNAs are processed and spliced differently, and present differences also in their sequences, precursor structures, evolutionary origins, and biogenesis mechanisms [22,23,24]. Plant miRNAs are methylated on the ribose of the 3′-end nucleotide, and methylation makes plant naked miRNAs highly stable relative to animal miRNAs [25], since most animal miRNAs are not methylated. It is believed the methylation at the 3′ ends by Hua Enhancer 1 (HEN1) distinguishes plant miRNAs from animal miRNAs [26]. The recent detection of plant miRNAs in body fluids suggests that these circulating miRNAs may have broad implications for miRNA-mediated gene expression control. These miRNAs could be associated with Ago2 [5], packaged into secreted ELNs [13] or exosomes [3], and thus be resistant to harsh internal environments, such as the acidic gastric environment, different temperature conditions, and nuclease enzymes [27]. This can provide favorable conditions for plant miRNA regulation in mammals.

MIR167e-5p is a conserved and highly expressed miRNA throughout the plant kingdom. In miRBase (http://www.mirbase.org/), there are 63 miRNAs sharing identical sequences (UGAAGCUGCCAGCAUGAUCUG). These miRNAs have been found in 27 plants, including soybean [28], wheat [29], and maize [30], which together account for a significant proportion of the diets of humans and other animals. Furthermore, MIR167e-5p has been found in *Moringa oleifera*, a widespread plant with substantial nutritional and medicinal value, and has a potential for cross-species control of human gene expression [31]. It was also found in human serum [32], human breast milk [33], porcine breast milk [34], and honey [9]. Maize MIR167e-5p was present in porcine nonsolid (blood) and solid tissues [35]. A study found that the basal level of MIR167a in human plasma was negligible but significantly increased after drinking watermelon juice, reaching 797.47 fM at the end point [36]. Ginger MIR167a downregulates the expression of SpaC in *Lactobacillus rhamnosus* in the mouse intestine [15]. These studies have proved that MIR167e-5p may be absorbed by mammals, may mediate cross-kingdom communication, and therefore may be a bioactive ingredient for epigenetic regulation in humans and animals.

Plant miRNAs are of potential biological function in mammalian cells. Plant MIR2911 has been reported to target the *PB2* and *NS1* genes and inhibit H1N1 viral replication [6]. MIR159 was capable of inhibiting cancer proliferation by targeting TCF7 [7]. MIR156a can directly target JAM-A and reduce inflammatory cytokine-induced monocyte adhesion [8]. Plant MIiR162a in beebread can even regulate honeybee caste development [9]. In our research, we proved that MIR167e-5p could inhibit the proliferation of IPEC-J2 and Caco-2 cells in vitro, as indicated by cell counting, MTT assay, and EdU incorporation assay. PCNA, which is a nuclear polypeptide expressed only in proliferating cells, has been taken as a good indicator of cell proliferation [37]. In the present study, PCNA mRNA level was significantly decreased by MIR167e-5p. These results provide sufficient evidence that MIR167e-5p can inhibit the proliferation of enterocytes.

β-catenin is a crucial component of the Wnt signaling pathway, which is crucial in many developmental processes [38], including the development of the intestinal epithelium [39,40,41]. In our research, plant MIR167e-5p could target β-catenin, as supported by bioinformatics analysis and luciferase report system experiments. qPCR and western blot further proved that β-catenin mRNA and protein levels were significantly decreased by MIR167e-5p mimics, while MIR167e-5p inhibitor relieved this inhibition. In previous studies, blockade of β-catenin or overexpression of the Wnt inhibitor Dkk-1 resulted in a severe loss of proliferative epithelial cells in the intestine [42]. Mutations in the negative regulator of Wnt signaling, the adenomatous polyposis coli (APC) protein, or overexpression of oncogenic forms of β-catenin resulted in hyperproliferation of the epithelium [43]. In our research, β-catenin mRNA levels recovered to the levels observed in NC control cells after treatment with MIR167e-5p inhibitor, but the protein levels were only mildly recovered in Caco-2 cells, similar to PCNA and c-Myc levels. The Wnt/β-catenin signaling pathway is abnormally activated in colon cancer [44]. Caco-2 cells differ from normal enterocytes, bearing mutations in the APC and β-catenin genes, which are involved in the transcriptional control of the myc gene during enterocyte differentiation. The ability of APC to regulate β-catenin levels in the cytosol and nucleus has been well demonstrated and involves binding of APC to β-catenin and stimulation of its phosphorylation and degradation [45]. Mutations or regulatory changes that disrupt APC or inhibit the APC-mediated β-catenin turnover may lead to altered levels of cytosolic and nuclear β-catenin. Some evidence has indicated that cell fate determination and lineage commitment of enterocytes are not governed by a simple rule, but by a complex and multilayered interaction between several pathways [46]. This means that β-catenin is regulated in many ways, for example, NOTCH1 plays a key role in the Wnt pathway, and activation of NOTCH1 is associated with the translocation of β-catenin to the nucleus in colon cancer [47]. Axin expression is positively regulated by β-catenin, and this generates a negative feedback loop that would be expected to decrease cytosolic β-catenin levels [48]. Besides, altering the level and distribution of β-catenin may affect the paralogous protein plakoglobin (γ-catenin) which, like β-catenin, can bind cadherins, APC, and TCFs, mimicking the effects of Wnt signaling [49]. The c-Myc transcription factor is a potent inducer of proliferation and is required for Wnt/β-catenin signaling in the intestinal epithelium. C-Myc appears essential to provide crypt progenitor cells with the necessary biosynthetic capacity to successfully progress through the cell cycle. C-Myc-deficient crypts are lost within weeks and replaced by c-Myc-proficient crypts through a fission process of crypts that have escaped gene deletion [50]. Considering with these observations, we can confirm that MIR167e-5p may inhibit enterocyte proliferation by targeting the β-catenin pathway.

As an essential organ for digestion and absorption, the intestine is the largest immune organ and plays a key role in maintaining animal health and growth. Recent studies have indicated that miRNAs can regulate gene expression in different species, and plant miRNAs regulate the intestinal flora and, thus, intestinal development. MiRNAs in EPDELNs have been detected and found to have the potential to regulate human mRNAs [14]. GELNs cause significant induction of Lgr5 intestinal stem cells through the Wnt/β-catenin pathway, protecting mice from DSS-induced colitis [12]. EPDELNs could induce the expression of genes encoding anti-inflammatory cytokines, anti-oxidation proteins, and proteins activating Wnt signaling [13]. GELNs mdo-miR7267-3p-mediated targeting of the LGG monooxygenase ycnE leads to increased indole-3-carboxaldehyde, inducing the production of IL-22 and thus ameliorating mouse colitis via IL-22-dependent mechanisms [15]. Our study suggests that plant MIR167e-5p might execute its inhibitory function in porcine and human enterocytes. These results demonstrate that plant miRNAs can regulate intestinal growth in vitro. Previous studies have been suggested that food-derived plant miRNAs may be important factors regulating intestinal health and development.

In conclusion, this study provides new evidence supporting the idea that plant miRNAs can exert cross-kingdom regulatory functions in mammals. We provide the first evidence that plant MIR167e-5p suppresses intestinal cell proliferation by targeting β-catenin. We suggest that the large variety of plant miRNAs should be considered as a new class of nutrients, which has been neglected for centuries.

## Figures and Tables

**Figure 1 cells-08-01385-f001:**
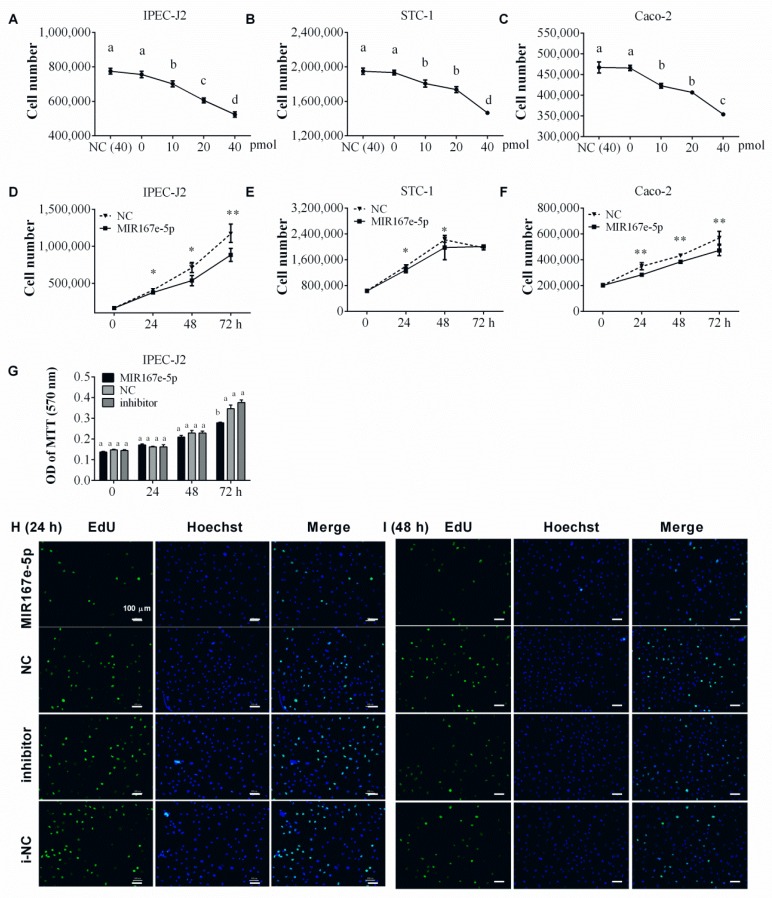
Synthetic MIR167e-5p suppresses the proliferation of IPEC-J2, STC-1, and Caco-2 cells. (**A**–**C**) Number of IPEC-J2 (**A**), STC-1 (**B**), and Caco-2 (**C**) cells transfected with different doses of MIR167e-5p (*n* = 6, LSD tests). (**D**–**F**) IPEC-J2 (**D**), STC-1 (**E**), and Caco-2 (**F**) cells proliferation after transfection with NC or MIR167e-5p mimics at 0, 24, 48, and 72 h. Cell counting was performed to evaluate cell proliferation (*n* = 6, *t*-tests). (**G**) Cell metabolic activity was examined by the MTT assay 0, 24, 48, and 72 h after transfection with MIR167e-5p, NC, inhibitor, or i-NC (*n* = 7, LSD tests). (**H**,**I**) Immunofluorescence staining for EdU incorporation in IPEC-J2 cells 24 and 48 h after treatment with MIR167e-5p, NC, inhibitor, or i-NC (Scale bars, 100 μm, *n* = 6); * *p* <0.05; ** *p* <0.01; the same superscripts denote a non-significant difference (*p* > 0.05), and different superscripts denote a significant difference (*p* < 0.05).

**Figure 2 cells-08-01385-f002:**
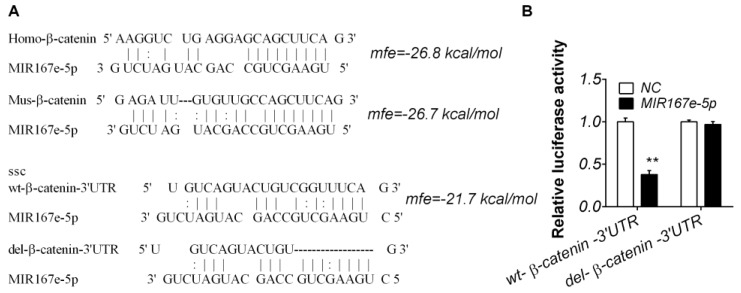
MIR167e-5p targets β-catenin. (**A**) The predicted target β-catenin 3′-UTR (untranslated region) and MIR167e-5p. Black lines and two-dotted labels represent the paired and G:U paired sequences for normal and 3′-UTR-deleted β-catenin, respectively. Note that the potential binding site of MIR167e-5p in the β-catenin 3′-UTR is highly conserved across species. (**B**) HeLa cells were transfected with each of the constructed plasmids, together with MIR167e-5p or NC. MIR167e-5p significantly decreased luciferase activity in cells transfected with wild-type (wt) β-catenin but not in cells transfected with 3′-UTR-deleted β-catenin (firefly luciferase activity/Renilla luciferase activity, ** *p* < 0.01, *n* = 8).

**Figure 3 cells-08-01385-f003:**
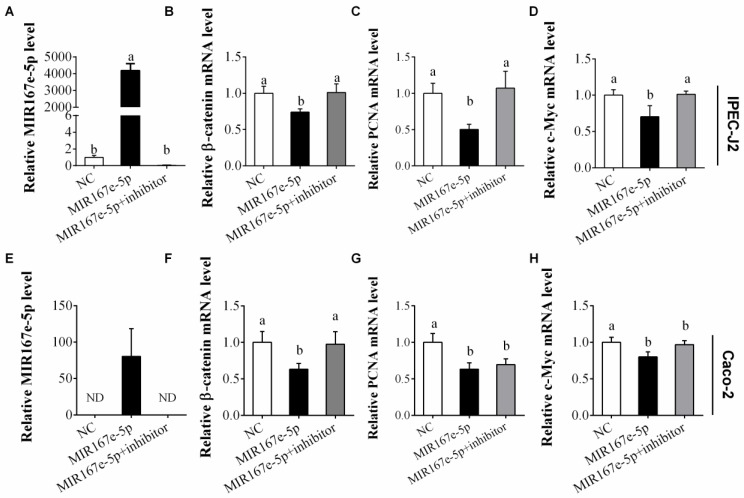
MIR167e-5p suppresses β-catenin, PCNA, and c-Myc mRNA expression. (**A****,E**) qPCR analysis of relative MIR167e-5p levels in IPEC-J2 and Caco-2 cells after transfection with NC, MIR167e-5p, or MIR167e-5p+inhibitor (normalized against U6; *n* = 6). (**B**–**D**) qPCR analysis of relative β-catenin (**B**), PCNA (**C**), and c-Myc (**D**) mRNA levels in IPEC-J2 cells after transfection with NC, MIR167e-5p, or MIR167e-5p+inhibitor (normalized against β-actin; *n* = 6). (**F**–**H**) qPCR analysis of relative β-catenin (**F**), PCNA (**G**), and c-Myc (**H**) mRNA levels in Caco-2 cells after transfection with NC, MIR167e-5p, or MIR167e-5p+inhibitor (normalized against β-actin; *n* = 6). Statistical significance was determined by LSD tests. The same superscripts denote non-significant differences (*p* > 0.05), different superscripts denote significant differences (*p* < 0.05).

**Figure 4 cells-08-01385-f004:**
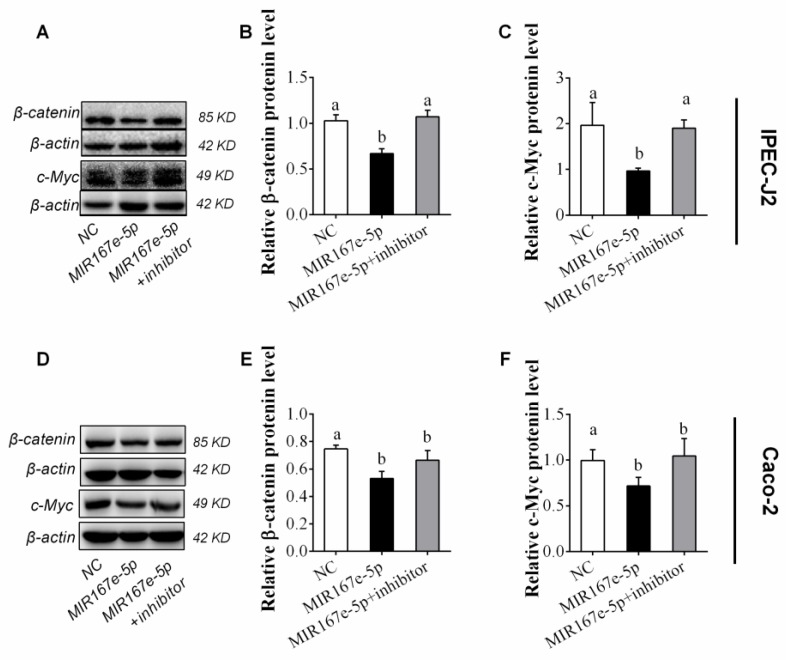
MIR167e-5p suppresses β-catenin and c-Myc protein expression. (**A**–**C**) Western blot was used to evaluate β-catenin (**A**,**B**) and c-Myc (**A**,**C**) protein levels in IPEC-J2 cells after transfection with NC, MIR167e-5p, or MIR167e-5p+inhibitor (*n* = 6; LSD tests). (**D**–**F**) β-catenin (**D**,**E**) and c-Myc (**D**,**F**) protein levels in Caco-2 cells after transfection with NC, MIR167e-5p, or MIR167e-5p+inhibitor (*n* = 6; LSD tests). The same superscripts denote non-significant differences (*p* > 0.05), different superscripts denote significant differences (*p* < 0.05).

**Table 1 cells-08-01385-t001:** qPCR and PCR primers.

Gen	Sequences	Product Length (bp)	TM	Gene Accession
Stem-loop-RT-MIR167e-5p primer	GTCGTATCCAGTGCGTGTCGTGGAGTCGGCAATTGCACTGGATACGAC CAGATC	74		
MIR167e-5p-F	TGAAGCTGCCAGCATGAT	68	56	
Stem-loop Universal	ATCCAGTGCGTGTCGTGGA			
Ssc-PCNA-F	TTCTTCCACCTGTAGCCG	269	60	NM_001291925.1
Ssc-PCNA-R	TTGGACATGCTGGTGAGG			
Ssc-c-Myc-F	GTCCAAGCAGAGGAGCAAA	103	58	HF549032.1
Ssc-c-Myc-R	ATGGGCAAGAGTTCCGTAG			
Ssc-β-catenin-F	TGAACCTGCCATCTGTGC	88	58	AB046171.1
Ssc-β-catenin-R	TCCGTAGTGAAGGCGAAC			
Homo-PCNA-F	AGGCACTCAAGGACCTCATC	250	58	CR541799.1
Homo-PCNA-R	GCCAAGGTATCCGCGTTATC			
Homo-β-catenin-F	CTGGCAGCAACAGTCTTACC	224	58	X87838.1
Homo-β-catenin-R	ACATAGCAGCTCGTACCCTC			
Homo-c-Myc-F	CACATCAGCACAACTACGCA	119	58	NM_002467.6
Homo-c-Myc-R	GGTGCATTTTCGGTTGTTGC			
β-actin-F	CCAGCACCATGAAGATCAAGATC	55	60	AY550069.1
β-actin-R	ACATCTGCTGGAAGGTGGACA			
U6-F	CTCGCTTCGGCAGCACA	71	60	NR_004394
U6-R	AACGCTTCACGAATTTGCGT

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
