# Peer review of "Plant MIR167e-5p Inhibits Enterocyte Proliferation by Targeting β-Catenin"

_cells, 2019, doi:10.3390/cells8111385_

Round 1
Reviewer 1 Report
Minor1: figure 2 luciferase is not fluorescence but chemiluminescence (light is being produced NOT emitted!) the y-ax should be tabled as "relative luciferase units" when not normalised to the blanc or relative luciferase activity as it s set to one to NC.
Major1:
line 240 remarkably recovered by the inhibitor pannel 3c and 3d sorry for 3c I agree though 3d this is not the case please adapt. please also adapt in the discussion as this strongly weakens the role of c-myc (line 310 plus introduction and introduction)
Minor2: check figure numbering i see 2 times figure 4...
Major2" (fig 4 line 258) and lines 251-256:
Western blotting is a technique for semi-quantitative detection of proteins therefore should not be quantified in this way but taken as it is. The quantification of the blots is generally considered to be unreliable, and to try quantification it cost a large amount of antibodies of consistent quality and testing them with ApoA-I standards, to fully optimize the Western blotting which does imply a methodological research paper in itself to obtain (1, 2). In short here the value of quantification is missing. If you want to do this it cost too much effort and also many repetitions of the experiment which is currently lacks body to allows statistics and quantification.
1. Charette, S. J., Lambert, H., Nadeau, P. J., and Landry, J. (2010) Protein quantification by chemiluminescent Western blotting: elimination of the antibody factor by dilution series and calibration curve. Journal of immunological methods 353: 148-150
2. Taylor, S. C., Berkelman, T., Yadav, G., and Hammond, M. (2013) A defined methodology for reliable quantification of Western blot data. Molecular biotechnology 55: 217-226
Quantification should not be needed if the effect is clearly there. Here the emphasize of the downregulation of B catenin and c-Myc seem statistically significant however this is definitely not clear if you look at the western blots. I would suggest to rewrite this part and NOT to quantify western blots. Write in terms like: MIR167e-5p seems to downregulate b-catenin. c-Myc might be downregulated too as the b-actin is slightly higher though this needs further investigation.
This also means the conclusions need to be a bit less strong (don’t use the words significant as statistics seems not applicable).
for instance line 254 50.92% is a huge decrease you should instantly see this in a blot. I don't se it. (I know eyes are tricky but its not the proper method as it not linear and not validated to be quantified)
Author Response
Dear Reviewer: Thank you for your comments concerning our manuscript entitled "MIR167e-5p inhibits proliferation of IPEC-J2 by targeting β-catenin" (ID: cells-532329). Those comments are all valuable are very helpful for revising and improving our paper, as well as the important guiding significance to our researches. We have studied comment carefully and have made correction which we hope meet with approval. Revised portion are mared in red in the paper. The main section in the paper and reponds to the comment are as flowing: Minor1: figure 2 luciferase is not fluorescence but chemiluminescence (light is being produced NOT emitted!) the y-ax should be tabled as "relative luciferase units" when not normalised to the blance or relative luciferase activity as its set to one to NC. Response minor1: Thanks for your advice, we have corrected the mistakes. Please see figure 2B. Major1: line 240 remarkably recovered by the inhibitor pannel 3c and 3d sorry for 3c I agree though 3d this is not the case please adapt. Please also adapt in the discussion as this strongly weakens the role of c-myc (line 310 plus introduction and introduction). Response major1: Thank you for your question. Probably, NC is higher than expected. Western blot results showed in figure 4C and D are clearer.We redid this experiment and rewrited this part, and please figure 3D and line 239-240. Minor2: check figure numbering i see 2 times figure 4... Response minor2:Thanks for your careful inspection and we are very sorry for that. We have corrected the mistakes and checked through the full manuscript. Major2" (fig 4 line 258) and lines 251-256: Western blotting is a technique for semi-quantitative detection of proteins therefore should not be quantified in this way but taken as it is. The quantification of the blots is generally considered to be unreliable, and to try quantification it cost a large amount of antibodies of consistent quality and testing them with ApoA-I standards, to fully optimize the Western blotting which does imply a methodological research paper in itself to obtain (1, 2). In short here the value of quantification is missing. If you want to do this it cost too much effort and also many repetitions of the experiment which is currently lacks body to allows statistics and quantification. 1. Charette, S. J., Lambert, H., Nadeau, P. J., and Landry, J. (2010) Protein quantification by chemiluminescent Western blotting: elimination of the antibody factor by dilution series and calibration curve. Journal of immunological methods 353: 148-150 2. Taylor, S. C., Berkelman, T., Yadav, G., and Hammond, M. (2013) A defined methodology for reliable quantification of Western blot data. Molecular biotechnology 55: 217-226 Quantification should not be needed if the effect is clearly there. Here the emphasize of the downregulation of B catenin and c-Myc seem statistically significant however this is definitely not clear if you look at the western blots. I would suggest to rewrite this part and NOT to quantify western blots. Write in terms like: MIR167e-5p seems to downregulate b-catenin. c-Myc might be downregulated too as the b-actin is slightly higher though this needs further investigation. This also means the conclusions need to be a bit less strong (don’t use the words significant as statistics seems not applicable). for instance line 254 50.92% is a huge decrease you should instantly see this in a blot. I don't se it. (I know eyes are tricky but its not the proper method as it not linear and not validated to be quantified) Response major2: Thanks for your advice. We have rewrited this part, and please see line252-256. We keep the quantification and statistics to show a more concrete result. Thank you and best regards. Yours sincerely, Meng Li
Reviewer 2 Report
In this study, the authors showed the potential of plant MIR167e-5p in the suppression of the proliferation of IPEC-J2 cells. The authors also showed that plant MIR167e-5p was able to down-regulate beta-catenin pathway targeting beta-catenin.
Experimental findings are presented in a clear manner and the demonstration of the inhibitory activity of the plant MIR167e-5p on beta-catenin is convincing.
However, the relevance of the study depends on the expression of the plant MIR167e-5p in vivo. The action of the plant MIR167e-5p could not be really relevant in vivo, because the concentration of the plant MIR167e-5p reached in animal cells could not be sufficient to achieve the inhibitory action on intestinal cells.
Major comments
1) Results presented in figure 4 should be obtained also in a human intestinal cell line.
2) Authors should discuss more extensively about the expression of plant MIR167e-5p. It is important to know if plant MIR167e-5p is highly expressed in some vegetables and if plant MIR167e-5p has been found in the blood of animals fed with vegetables containing an elevated amount of plant MIR167e-5p. This point is crucial to establish the importance of plant MIR167e-5p as nutraceutical compound.
3) The experiment showed in figure 3D should be repeated. Probably, NC is higher than expected. Western blot results showed in figure 4C and D are clearer.
Minor comments
1) In figure 1A, is shown that the concentration of 2'-O-methylated plant MIR167e-5p that achieved the highest effect on cell proliferation inhibition is equal to 40 pmol. The authors should explain why they used 20 pmol in other experiments.
2) In the legend of figure 1, there are two D letters to indicate C and D panels.
3) There are two figure 4, the first one should be labeled as figure 3.
4) In figure 3 B, C, and D panels (mislabeled as figure 4) authors should put the reference sample (NC) equal to one. In the text (lines 232-234), it is not clear which panel of the figure shows PCNA results.
Author Response
Dear Reviewer: Thank you for your comments concerning our manuscript entitled "MIR167e-5p inhibits proliferation of IPEC-J2 by targeting β-catenin" (ID: cells-532329). Those comments are all valuable are very helpful for revising and improving our paper, as well as the important guiding significance to our researches. We have studied comment carefully and have made correction which we hope meet with approval. Revised portion are mared in red in the paper. The main section in the paper and reponds to the comment are as flowing: Major comments 1) Results presented in figure 4 should be obtained also in a human intestinal cell line. Response 1: Thanks for your advice, we very sorry for that we did not get the unavailable human intestinal epithelial cell line to perform the experiment. 2) Authors should discuss more extensively about the expression of plant MIR167e-5p. It is important to know if plant MIR167e-5p is highly expressed in some vegetables and if plant MIR167e-5p has been found in the blood of animals fed with vegetables containing an elevated amount of plant MIR167e-5p. This point is crucial to establish the importance of plant MIR167e-5p as nutraceutical compound. Response 2: Thanks for your advice, we have supplemented more extensive discussion about the expression of plant, and please see line 276-288. 3) The experiment showed in figure 3D should be repeated. Probably, NC is higher than expected. Western blot results showed in figure 4C and D are clearer. Response 3: Thanks for your advice, we redid this experiment and new figure is provided, and please figure 3D. Minor comments 1) In figure 1A, is shown that the concentration of 2'-O-methylated plant MIR167e-5p that achieved the highest effect on cell proliferation inhibition is equal to 40 pmol. The authors should explain why they used 20 pmol in other experiments. Response 1: Thank you for your question. The results showed in figure 1A, 10 pmol enough to inhibit the cell proliferation, 20 pmol significantly inhibit the cell proliferation. We used 20 pmol as test dose, to ensure effective concentration and it doesn't have to be a highest effect. 2) In the legend of figure 1, there are two D letters to indicate C and D panels. Response 2:Thanks for your careful inspection and we are very sorry for that. We have corrected the mistakes and checked through the full manuscript. 3) There are two figure 4, the first one should be labeled as figure 3. Response 3:Thanks for your careful inspection and we are very sorry for that. We have corrected the mistakes and checked through the full manuscript. 4) In figure 3 B, C, and D panels (mislabeled as figure 4) authors should put the reference sample (NC) equal to one. In the text (lines 232-234), it is not clear which panel of the figure shows PCNA results. Response 4: Thanks for your advice. We have corrected the mistakes and put the reference sample (NC) equal to one. Please figure 3B, C and D. Thank you and best regards. Yours sincerely, Meng Li
Round 2
Reviewer 1 Report
no more comments
Author Response
Response: We acknowledge the your suggestion very much, which are valuable in promoting the study of this filed.
Reviewer 2 Report
The authors significantly improved the quality of the manuscript.
Author Response

(The authors gave the same response as above.)
